biochemistry, behaviour, evolution

dogs, palaeofaeces, palaeoproteomics, zooms, Nunalleq Alaska, archaeology

**Authors for correspondence:**
Anne Kathrine W. Runge
e-mail: ak@palaeome.org
Camilla Speller
e-mail: camilla.speller@york.ac.uk

# Palaeoproteomic analyses of dog palaeofaeces reveal a preserved dietary and host digestive proteome

Anne Kathrine W. Runge[1,2], Jessica Hendy[1,3], Kristine K. Richter[4,5], Edouard Masson-MacLean[6], Kate Britton[6,7], Meaghan Mackie[2,8], Krista McGrath[1,9], Matthew Collins[2,10], Enrico Cappellini[2] and Camilla Speller[1,11]

[1]BioArCh, Department of Archaeology, University of York, Environment Building, Wentworth Way, YO10 5DD York, UK
[2]Section for Evolutionary Genomics, the GLOBE Institute, University of Copenhagen, Øster Farimagsgade 5A, 1353 København K, Denmark
[3]Department of Archaeogenetics, and [4]Department of Archaeology, Max Planck Institute for the Science of Human History, Kahlaische Strasse 10, 07743 Jena, Germany
[5]Department of Anthropology, Harvard University, Cambridge, MA 02138, USA
[6]Department of Archaeology, University of Aberdeen, Aberdeen, Scotland, UK
[7]Department of Human Evolution, Max Planck Institute for Evolutionary Anthropology, Deutscher Platz 6, Leipzig 04103
[8]The Novo Nordisk Foundation Center for Protein Research, University of Copenhagen, Blegdamsvej 3b, 2200 København N, Denmark
[9]Department of Prehistory and Institute of Environmental Science and Technology (ICTA), Universitat Autònoma de Barcelona, 08193 Bellaterra, Spain
[10]Department of Archaeology, University of Cambridge, Cambridge CB2 3DZ, UK
[11]Department of Anthropology, University of British Columbia, 6303 NW Marine Drive, Vancouver, Canada V6T 1Z1

AKWR, 0000-0003-2421-4831; KKR, 0000-0003-3591-6900; MC, 0000-0003-4226-5501; CS, 0000-0001-7128-9903

The domestic dog has inhabited the anthropogenic niche for at least 15 000 years, but despite their impact on human strategies, the lives of dogs and their interactions with humans have only recently become a subject of interest to archaeologists. In the Arctic, dogs rely exclusively on humans for food during the winter, and while stable isotope analyses have revealed dietary similarities at some sites, deciphering the details of provisioning strategies have been challenging. In this study, we apply zooarchaeology by mass spectrometry (ZooMS) and liquid chromatography tandem mass spectrometry to dog palaeofaeces to investigate protein preservation in this highly degradable material and obtain information about the diet of domestic dogs at the Nunalleq site, Alaska. We identify a suite of digestive and metabolic proteins from the host species, demonstrating the utility of this material as a novel and viable substrate for the recovery of gastrointestinal proteomes. The recovered proteins revealed that the Nunalleq dogs consumed a range of Pacific salmon species (coho, chum, chinook and sockeye) and that the consumed tissues derived from muscle and bone tissues as well as roe and guts. Overall, the study demonstrated the viability of permafrost-preserved palaeofaeces as a unique source of host and dietary proteomes.

## 1. Introduction

The domestic dog (*Canis lupus familiaris*) originated in Eurasia [1,2] at least 15 000 years ago [3], but despite our extended collaborative history, the lives of dogs and their management through time has largely been neglected by

researchers [4]. However, attitudes towards the lived experiences of dogs and their roles in past human societies are beginning to shift [5], and in recent years, dietary studies have provided novel insight into their lifeways and how different cultures incorporated dogs into their subsistence strategies. Dog and human diets can be so similar that they can be analysed in lieu of humans as a proxy for reconstructing human diet [6] and have revealed a reliance on similar resources [7] although comparative studies indicate that this is not always the case [8]. Genetic analyses have further shown that as human subsistence practices changed with the spread of agriculture, the diet of non-arctic dogs was impacted as well [9].

The earliest known dog remains in North America indicate that this domesticated species was introduced at least 10 000 years ago and belonged to a now extinct lineage of arctic dog [10,11]. In the North American Arctic, dogs do not appear to have been present in large numbers until around 1000 BP [12]. The ancestors of Yup'ik, Inuit and other Indigenous Arctic groups relied on advanced transportation technology including dog traction [13] and appear to have been the first to introduce specialized sled dogs to North America [14]. Arctic dogs rely exclusively on humans for food during the long winters, but may have been fed differently or less frequently in summer, or let loose to fend for themselves [15,16]. Working sled dogs are a particularly expensive resource, requiring up to 3.2 kg of fish or meat every day [17] and provisioning of dogs would therefore have played a significant role in the food procurement strategies of past arctic cultures. Stable isotope analyses from coastal sites in Alaska [7,18,19] and Canada [20–22] have shown that the majority of dog diet in ancient and historical North American Arctic cultures consisted of marine mammals and salmonids. These studies provide important insight into the diet of dogs and humans, revealing similarities in resource allocation, but do not provide information on seasonal variation unless incrementally developed tissues such as hair or claw are targeted [7]. In addition, the practice of feeding specific species, such as walrus [17] or chum salmon (also known as 'dog salmon') [7], or animal parts, including hide, that were considered unpalatable or unsuitable for human consumption can be difficult to characterize in any detail using stable isotope approaches.

Recent advances in the field of palaeoproteomics have opened up new avenues for dietary studies of past populations, complementing more traditional zooarchaeology, palaeobotany and stable isotope approaches. Proteins are frequently tissue specific, allowing distinct plant parts (e.g. seeds, leaves and roots) and animal products (e.g. muscle, milk and blood) to be distinguished [23]. In particular, ancient dental calculus has emerged as a valuable substrate for preserving dietary biomolecules, revealing novel insights especially in ancient dairy consumption [24,25]. Palaeoproteomics has also been used to identify the 'last meal' of Ötzi, a human Copper Age mummy [26] and to reveal evidence of breast milk feeding from the rib bone of a neonatal dog [27], as well as food composition in a range of well-preserved archaeological materials [28–30]. Staining techniques have previously demonstrated the survival of proteins in palaeofaeces [31], a material which has long been used for macro- and microscopic dietary analyses [32]. However, palaeoproteomic analyses have only been applied to the substrate with limited success [33]. Ancient DNA evidence has shown palaeofaeces to preserve dietary

information as well as evidence of the gut microbiome [34,35], but the scope of these studies has been limited owing to its complex and highly degradable nature [32]. Furthermore, DNA studies are limited by a lack of tissue specificity, which is a major strength of palaeoproteomics. Here, in order to investigate the preservation of ancient proteins in palaeofaeces, and to reveal new insights into the subsistence of arctic dogs, we use shotgun palaeoproteomics to identify a range of host and dietary-derived proteins preserved in palaeofaeces and complement this analysis with zooarchaeology by mass spectrometry (ZooMS) on bone fragments collected from within this substrate.

## 2. Material and methods

### (a) Samples
All samples were excavated from the Nunalleq site, southwest Alaska (figure 1) during research excavations in 2013–2015. The site, a pre-contact Yup'ik village close to the contemporary community at Quinhagak, Alaska, was occupied between *ca* 1300 CE and 1750 CE. It has exceptional preservation of archaeological artefacts and bioarchaeological remains [36], including dog harnesses made of braided grass, dog bones, fur and even lice, as well as palaeofaeces [16]. Detailed site and sample information can be found in the electronic supplementary material, file S1.

### (b) Palaeoproteomics
Five samples (figure 1; electronic supplementary material, table S1) were selected for protein analysis. Palaeoproteomic extraction was carried out following the protocol of Tsutaya *et al.* [37] designed for extraction of modern faeces. Full methodological details on protein extraction and data analysis can be found in the electronic supplementary material, file S1. Samples were analysed using liquid chromatography tandem mass spectrometry (LC-MS/MS) and spectral data were converted to Mascot generic format (.mgf). Semi-tryptic MS/MS ion database searching was performed on Mascot (Matrix ScienceTM, v. 2.6.1) against a database composed of Swiss-Prot, and the *Canis lupus familiaris* and *Oncorhynchus mykiss* proteomes. Only proteins with a minimum of two non-identical peptides and those retained after an false discovery rate (FDR) estimation of 1% of distinct peptide spectrum matches were considered in further downstream interpretation. Keratins are considered separately (electronic supplementary material, table S3), while peptides identified in the extraction negative were excluded from further analysis along with matches to the reagents trypsin and lysyl peptidase (electronic supplementary material, table S4). Lowest common ancestor analysis was performed using Unipept metaproteomics analysis [38]. The taxonomy of peptides unclassified by Unipept were explored using protein–protein BLAST against all non-redundant protein sequences. STRING v. 11.0 [39] was used to analyse protein–protein interactions and proteome function using gene ontology (GO) annotations (electronic supplementary material, table S5), and expression profiles using human homologues was explored using the GeneCards suite [40].

### (c) Zooarchaeology by mass spectrometry
The samples Nun-2 and Nun-4 each contained five bone fragments, which were analysed using ZooMS. Two pretreatment methods involving either acid demineralization (method A) or an ammonium bicarbonate buffer (method B) were used prior to collagen extraction. Samples were processed according to previously established protocols [41,42] and analysed on a MALDI-TOF mass spectrometer. MALDI spectra were processed using the R package MALDIquant [43] and visualized in mMass [44].

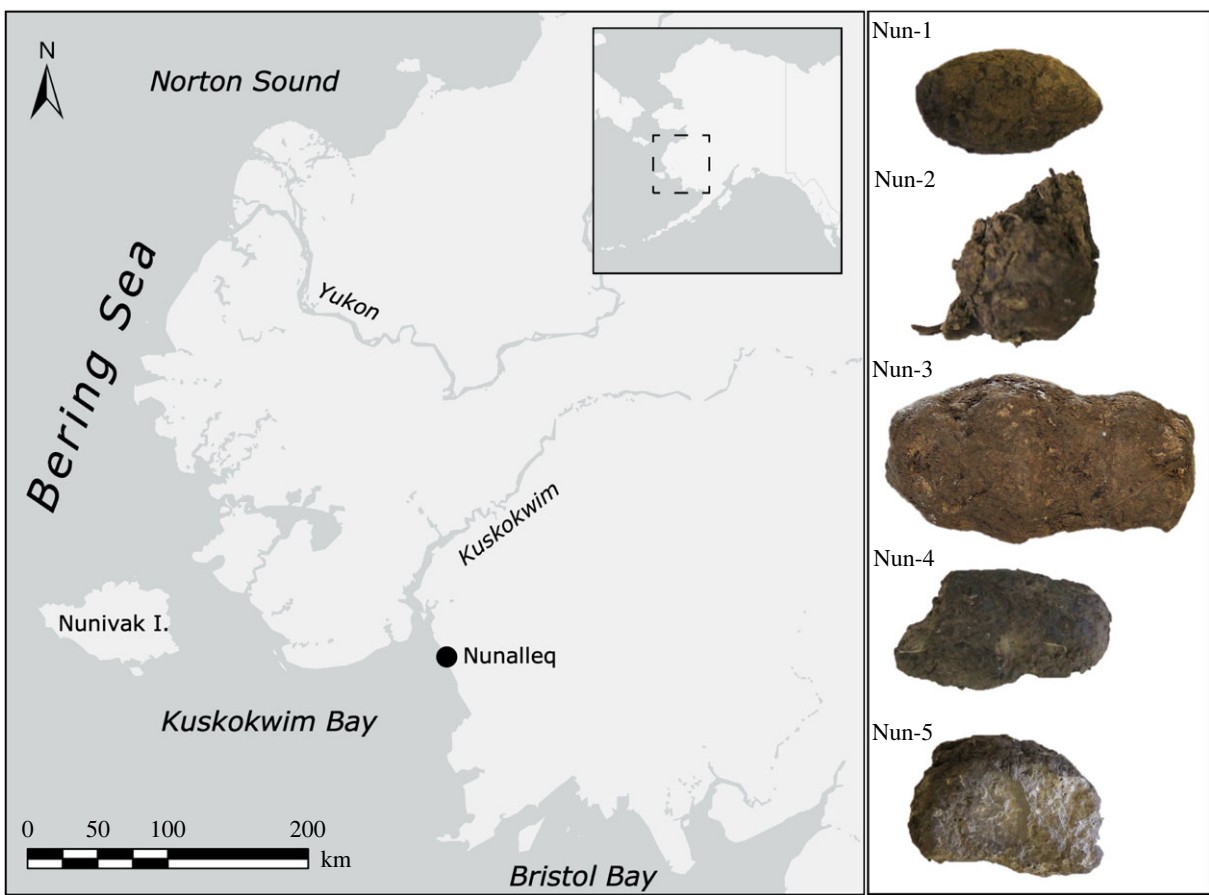

**Figure 1.** Map showing the location of the Nunalleq site in Alaska and pictures of the palaeofaeces analysed in this study.

Identifications (electronic supplementary material, table S6) were made based on published markers for mammals [41,45–48] (electronic supplementary material, table S7) and fishes including Pacific salmonids [49] (electronic supplementary material, tables S8 and S9). Full methodological details can be found in the electronic supplementary material.

## 3. Results and discussion

### (a) Metaproteomics

We applied shotgun proteomic analysis to five palaeofaeces, revealing evidence of a digestive proteome as well as indicators of consumed food. After quality filtering, we detected a total of 83 proteins across all five samples, with 56 distinct individual proteins represented by 282 unique peptide sequences (excluding keratins) (electronic supplementary material, table S2; figure 2). Protein recovery varied between samples (figure 3), with Nun-3 showing evidence of 38 identified proteins and Nun-5 limited to two (electronic supplementary material, tables S1 and S2). All samples, except Nun-5, displayed proteins that could be classified as host-derived (table 1) and dietary (electronic supplementary material, table S10). In this study, the extraction method was selected to enhance the detection of dietary proteins by pelleting bacteria and other particulates present in the samples. As expected, we did not detect abundant microbial proteins related to the gut or faecal microbiome, although a few bacterial peptides were identified despite this methodological approach.

Four samples, Nun-1, Nun-2, Nun-3 and Nun-4, were excavated from house floor contexts and displayed superior protein preservation compared to Nun-5, which was excavated from a debris context (electronic supplementary material, table S1). It is likely that the archaeological context influenced biomolecular preservation of the palaeofaeces, with house floors offering protection that debris contexts did not. However, given the absence of potential proteins in Nun-5 and considering that the sample did not contain hair or bone fragments, it is possible that it was misidentified as palaeofaeces. As a result, this sample is not considered in further analyses.

### (b) Host proteins

#### (i) Host identification

First, we characterized the palaeofaecal proteome derived from the putative host. In accordance with the initial morphological characterization, we identified proteins associated with the gastrointestinal (GI) system, which were specific to the taxonomic orders of *Canis lupus familiaris* in all four successful samples. Further proteins were assigned to the clades Canidae, Caniformia and Carnivora. As some proteins are conserved across taxonomic units, proteins assigned to these clades were considered to have originated from dogs (table 1). Only a single protein assigned to Hominidae, neutrophil defensin 1, was identified in the dataset. Given the abundance of proteins derived from Carnivora, especially proteins associated with the digestive system, the combined evidence points to palaeofaeces deposited by canids.

#### (ii) Digestive proteins

LC-MS/MS analysis revealed evidence of a suite of proteins associated with the GI system (table 1), including digestion. These included, but were not limited to, colipase and inactive

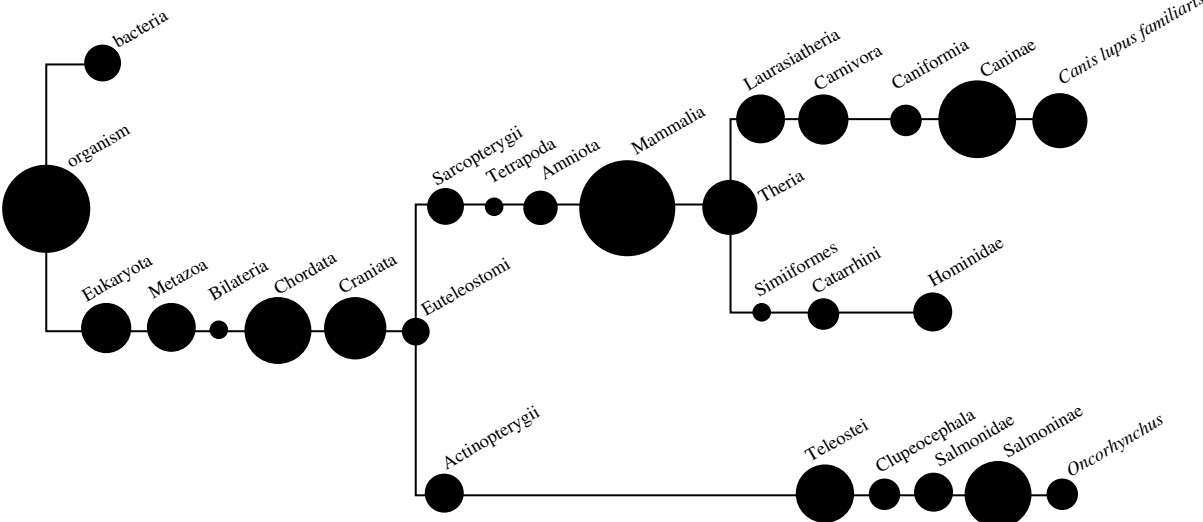

**Figure 2.** A schematic diagram of a cladogram indicating taxonomic distribution of peptides extracted from palaeofaeces across the five samples analysed from Nunalleq. The size of each bubble is proportional to the number of peptides assigned to each level. Taxonomic distribution was determined using Unipept meta-proteome analysis and further interrogated using BLAST against all non-redundant sequences. Keratins and laboratory contaminants are excluded from this schematic. The length of each branch does not represent evolutionary distance.

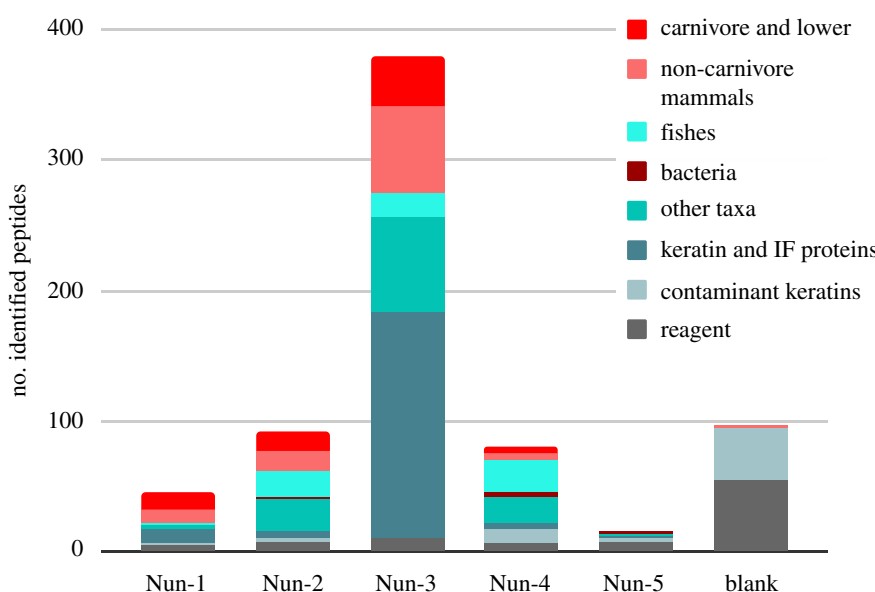

**Figure 3.** Number of identified peptides from each sample, grouped by broad taxonomy. The category carnivore and lower include all peptides assigned to Carnivora, Caniformia, Caninae and *Canis lupus familiaris*. Other taxa include all detected taxa except for mammals, fishes and bacteria. Peptides belonging to keratins and other intermediate filament (IF) proteins are classified separately, with contaminant keratins denoting those identified in the extraction blank. Post-translational modifications are not included in the peptide counts. (Online version in colour.)

pancreatic lipase-related protein 1 (involved in fat digestion), trefoil factor 2 (found in GI mucus layers) and aminopeptidase N (involved in protein digestion). In addition, proteins associated with the extracellular matrix, such as collagen (alpha-2(I)), were also detected. We note that some proteins, such as colipase, trefoil factor 2 and zymogen granule protein 16, were observed across all four samples, whereas others, such as aminopeptidase N, and Alpha 2-HS glycoprotein were only detected in individual samples (table 1). Serum albumin, a protein expressed in multiple tissues, was detected in Nun-3. Although bovine serum albumin is a common reagent in molecular biology (and in our study was detected in the negative control), the fact that the Nun-3 peptides match specifically to Carnivora, suggests that this is indeed an endogenous protein.

In order to examine the extent of biological interaction between the identified host proteins, we analysed protein–protein interactions (electronic supplementary material, figure S1) and found that the majority were connected within a network. Biological processes (electronic supplementary material, table S5) with the highest confidence included several relating to digestion and metabolism, such as digestion (GO:0007586, FDR 8.45E-07), primary metabolic process (GO:0044238, FDR 0.00023), organic substance metabolic process (GO:0071704, FDR 3.20E-03) and response to nutrient levels (GO:0031667, FDR 2.10E-03) (electronic supplementary material, table S5). Identified proteins also matched to protein (cfa04974, FDR 6.12e-07) and fat (cfa04975, FDR 3.90E-03) digestion and absorption Kyoto Encyclopedia of Genes and Genomes (KEGG) pathways

**Table 1.** Proteins assigned to the taxonomic level of carnivora or lower. (The taxonomic assignment of the protein accession is based on the peptide with the most specific taxonomy detected within a sample. Gene names are those associated with the Uniprot accession. Expression data was obtained from human homologues in GeneCards.)

| sample | identified host protein | Uniprot accession | gene name | most specific taxonomy | Mascot score | expression |
|---|---|---|---|---|---|---|
| Nun-1 | aminopeptidase N | A0A5F4CJ36[a] | ANPEP | Canidae | 84 | PJ, SI |
| | colipase | P19090[a] | CLPS | Caniformia | 67 | PJ |
| | deleted in malignant brain tumours 1 | A0A5F4BZQ1 | DMBT1 | *Canis lupus familiaris* | 76 | SI, C, PJ |
| | dipeptidyl peptidase 4 | F1PP08[a] | DPP4 | Canidae | 97 | MT incl. SI |
| | serpin family B member 6 | E2RGP2 | SERPINB6 | *Canis lupus familiaris* | 52 | MT incl. PJ SI, C |
| | trefoil factor 2 | F1PPU3[a] | TFF2 | Canidae | 90 | PJ, S |
| | zymogen granule protein 16 | E2RS34 | ZG16 | Canidae | 161 | SI, C, R |
| Nun-2 | anionic trypsin | P06872[a] | PRSS2 | *Canis lupus familiaris* | 98 | PJ, SI |
| | colipase | P19090[a] | CLPS | *Canis lupus familiaris* | 69 | PJ |
| | peptidase S1 domain-containing protein | F1PCE8 | LOC475521 | Canidae | 81 | N/A |
| | serpin family B member 6 | E2RGP2 | SERPINB6 | *Canis lupus familiaris* | 185 | MT incl. PJ SI, C |
| | syncollin | F1PWB9 | SYCN | *Canis lupus familiaris* | 71 | PJ, P |
| | trefoil factor 2 | F1PPU3[a] | TFF2 | Canidae | 64 | PJ, S |
| | zymogen granule protein 16 | E2RS34 | ZG16 | Canidae | 103 | SI, C, R |
| Nun-3 | alpha 2-HS glycoprotein | E2QUV3 | AHSG | Canidae | 108 | MT incl. C, P, PJ, R |
| | chymotrypsin like | F1PA60 | CTRL | Canidae | 53 | PJ, P |
| | colipase | P19090[a] | CLPS | *Canis lupus familiaris* | 345 | PJ |
| | collagen alpha-2(I) chain | F1PHY1 | COL1A2 | Canidae | 344 | MT incl. SI, C, R |
| | dipeptidyl peptidase 4 | F1PP08[a] | DPP4 | Canidae | 66 | MT incl. SI |
| | GLOBIN domain-containing protein | E2RLH6 | LOC609402 | Caniformia | 193 | MT incl. PJ, C, R |
| | | J9JHF7 | LOC100855540 | Caninae | 86 | N/A |
| | inactive pancreatic lipase-related protein 1 | A0A5F4CND7 | PNLIP | *Canis lupus familiaris* | 109 | PJ |
| | peptidase S1 domain-containing protein | A0A5F4CBC4 | CTRB2 | *Canis lupus familiaris* | 34 | PJ, P, R, gut (fetal) |
| | | A0A5F4CGL2 | KLK1 | *Canis lupus familiaris* | 80 | PJ, C, R |
| | | A0A5F4DFJ6[a] | PRSS2 | *Canis lupus familiaris* | 65 | PJ |
| | | F1PCE8 | LOC475521 | Canidae | 89 | N/A |
| | | F6XMJ9 | LOC478220 | *Canis lupus familiaris* | 91 | N/A |
| | serpin family B member 6 | E2RGP2 | SERPINB6 | *Canis lupus familiaris* | 117 | MT incl. PJ SI, C |
| | serpin family F member 1 | A0A5F4CK29 | SERPINF1 | Canidae | 86 | MT incl. C, P, R, S |
| | serum albumin | P49064 | ALB | Carnivora | 102 | MT incl. PJ, C, R |
| | trefoil factor 2 | F1PPU3[a] | TFF2 | Canidae | 70 | PJ, S |
| | zymogen granule protein 16 | E2RS34 | ZG16 | Canidae | 214 | SI, C, R |
| Nun-4 | colipase | P19090[a] | CLPS | *Canis lupus familiaris* | 49 | PJ |
| | trefoil factor 2 | F1PPU3[a] | TFF2 | Canidae | 51 | PJ, S |
| | zymogen granule protein 16 | E2RS34 | ZG16 | Canidae | 211 | SI, C, R |

[a]Denotes protein accessions involved in digestive and metabolic functions based on GO annotations. Expression abbreviations explained: B, blood; C, colon; L, liver; MT, multiple tissues; P, pancreas; PJ, pancreatic juice; R, rectum; S, stomach; SI, small intestine.

(electronic supplementary material, figure S1). Although trefoil factor 2 is among the GO terms assigned to digestion (electronic supplementary material, table S5), the STRING analysis (electronic supplementary material, figure S1) does not identify interactions with other host proteins identified here.

Digestion is facilitated by the secretion of fluids from the salivary glands, stomach, pancreas and small intestine [50]. Using data from UniProtKB and GeneCards, we also explored the tissues from which the identified host proteins were expressed using human protein homologues where expression data is better characterized. With the exception of four proteins for which no expression data was available, we found that all detected host proteins are present in pancreatic juice, small intestine, colon, rectum and stomach, with particularly high numbers expressed in human pancreatic juice, and the colon (table 1). Pancreatic juice neutralizes stomach acid as it passes from the stomach into the small intestine, and contains multiple digestive enzymes [51]. The detection of proteins expressed in the stomach, pancreatic juice, and the small intestine was therefore consistent with a digestive proteome, while proteins expressed in the colon and rectum were expected in a faecal proteome and supported a host origin. Together, these results indicate the potential for GI proteomes to be preserved within palaeofaeces. Future characterization of expression profiles of dog proteins will improve on these findings. Differences in protein expression between dogs and humans have been observed including in saliva [52], while variation may also exist between different dog lineages. For example, some non-arctic dogs display increased copy numbers of the AMY2B gene responsible for the production of amylase, the primary enzyme involved in starch digestion [53].

### (iii) Keratins

Keratins are ubiquitous contaminants in shotgun proteomic studies, often derived from handling or the presence of woollen fibres in the sampling or laboratory environment [54], which is exemplified in a previous study of palaeofaeces which detected primarily human keratins [32]. As a result, keratins are frequently excluded from further analysis. Nevertheless, our analysis revealed the presence of keratins and other intermediate filament (IF) proteins which could be attributed to *Canis lupus familiaris* in Nun-3, and to Canidae in Nun-1 and Nun-2 (electronic supplementary material, table S3). Peptides assigned to these proteins show evidence of deamidation, while the presence of 'pacman peptides' (peptides with fragmentation at multiple non-tryptic sites) may also indicate protein diagenesis consistent with ancient proteins (but see [55]). Keratins are present in intestinal epithelial cells [56] and are therefore likely to end up in faecal samples as ingested material passes through the digestive tract. In addition, hair was macroscopically observed in the palaeofaeces during sampling and may have originated either from the host or another dog, most likely ingested in relation to grooming. Finally, we note that keratins and IF proteins are potentially informative for dietary reconstruction from dog palaeofaeces as dogs are commonly fed on scraps, such as animal hide, not suitable for human consumption. At present, however, we do not have a secure strategy for distinguishing putative contaminant keratins from potential endogenous proteins that may be informative for diet.

### (c) Dietary proteins

Shotgun proteomic data revealed the presence of numerous proteins likely to originate from consumed food. Food-derived proteins were dominated by fishes, and the majority of these were conserved to the level of Salmoninae (electronic supplementary material, table S10). The majority of detected fish proteins derived from muscle tissue, including myosin motor domain-containing proteins, which could be assigned to the level of Salmoninae and Teleostei in Nun-3 and Nun-4, respectively. Additional fish-derived proteins, which were homologous to actin and titin, both involved in muscle contraction [52], were also detected. Type I collagen (alpha-1(I) and/or alpha-2(I) chains)—a major structural component of connective tissues including bone, tendons and skin—was identified in Nun-1, Nun-2 and Nun-4, and in all cases could be assigned to Salmoninae. Although type I collagen can be used to obtain species-level information through ZooMS and/or LC-MS/MS analysis of bone material, the coverage obtained in the palaeo-faeces LC-MS/MS dataset was not sufficient to assign these to a level below Salmoninae.

We identified fish vitellogenin, an egg storage protein surprisingly common in ancient samples [53–56], in Nun-2, Nun-3 and Nun-4. We also detected alpha-1-antiproteinase-like protein from Salmoninae in Nun-2 and Nun-4, and alpha-1,4 glucan phosphorylase from Teleostei in Nun-4. These two proteins are digestive enzymes, which probably originated from the consumption of fish guts. At arctic and subarctic sites like Nunalleq, successful occupation depended on the effective storage of resources that were seasonally available. While bone and muscle proteins may have derived from stored fishes, the presence of proteins associated with fish intestines suggests that some of the analysed palaeofaeces were deposited during the seasonal salmon run. During this time, caught salmon would have been gutted prior to being prepared for human consumption or storage. This is consistent with ethnographic accounts from northern Alaska, which have documented that dogs commonly ate intestines and roe in the summer [57].

These findings by LC-MS/MS analysis are complemented by ZooMS analysis applied to small bone fragments found within some of the palaeofaeces. Collagen spectra were obtained from five bone fragments from Nun-2 and two fragments from Nun-4 (electronic supplementary material, tables S6–S9). Two fragments from Nun-4 dissolved during pretreatment, while another was found to be misidentified and was probably a piece of wood. Two methods of pretreatment using acid demineralisation (method A) and ammonium bicarbonate buffer (AmBic, method B) were applied. Method A generated clearer spectra than method B (electronic supplementary material, tables S6–S9) suggesting that demineralization with acid to release a larger fraction of preserved collagen is preferable in cases where the bones have gone through a digestive tract prior to ZooMS and thus may be more poorly preserved. The bone fragments recovered from Nun-2 were identified as belonging to coho (*Oncorhynchus kisutch*), chinook (*Oncorhynchus tshawytscha*) and chum salmon (*Oncorhynchus keta*) (electronic supplementary material, table S6, tables S8 and S9). From Nun-4, one bone fragment was identified as sockeye salmon (*Oncorhynchus nerka*), and the last assigned to Canidae (electronic supplementary material, tables S6 and S7). These results further demonstrate that ZooMS analyses can successfully be applied to semi-digested bone fragments which are frequently preserved in dog and human palaeofaeces [58] and may provide complementary insight into diet [42]. While ZooMS can provide species-level information for Pacific salmonids, higher resolution analyses, such as LC-MS/MS may be required for this level of taxonomic resolution for mammal species [59].

Previously, indirect dietary evidence from zooarchaeological [7,16] and lipid residue analyses [60] as well as direct dietary evidence obtained from stable isotope analyses from humans [36,61,62] and dogs [7], have indicated a mixed economy at Nunalleq relying predominantly on salmonids, but also marine mammals and terrestrial herbivores. Even with a limited number of analysed samples, the ZooMS and LC-MS/MS analyses corroborated the reliance on salmonids as dog food. This consisted of a range of available salmon species, which may have been fed to dogs without distinction based on the multiple species identified in the Nun-2 sample (electronic supplementary material, table S6). Our findings therefore suggest that the reliance on chum salmon, the so-called 'dog salmon', as dog food, as previously suggested by local traditional knowledge [7], is a later adaptation, potentially related to the commercial value of the species. Interestingly, one bone fragment from Nun-4 was identified as canid, indicating that Nunalleq dogs chewed the bones of wolves, foxes or other dogs, supporting previous observations of dog gnaw marks on discarded dog bones [16]. As tissues from other canids (and especially canid bones) may form part of the diet of domestic dogs, observations of proteins assigned as 'host' may also result from dietary sources. In particular, collagen type I and chondroadherin are abundant in bone and cartilaginous tissues, respectively, and thus, identified peptides from these proteins may also derive from dietary sources, in addition to the host.

Ingested material normally passes through the digestive tract in a matter of days, and as a result, it is highly unlikely that every dietary source of the Nunalleq dogs has been detected here, especially considering the highly seasonal environment [16]. In the future, a more comprehensive study involving additional archaeological contexts and sites could provide important insight into the diversity of feeding strategies across arctic contexts, and beyond.

## (d) Challenges and future directions

This study demonstrates the use of palaeoproteomics in identifying the host species of palaeofaeces in cases where morphology is ambiguous, and further showcases the successful sequence identification of components of the host GI proteome. We also confirm that palaeoproteomics and ZooMS can provide high-resolution insight into short-term dietary intake. Nevertheless, given the novelty of this approach, there are several challenges and complexities to be considered in this and future studies.

### (i) Imprecise taxonomy and database biases

First, the highly conserved nature of some proteins makes it impossible to assign them to precise taxonomic units (electronic supplementary material, table S2), and differentiating between host proteins and elements of diet can be challenging. Thus, proteins assigned to mammalia or a higher taxonomic classification could derive from either the diet or the dog host. For example, mammalian haemoglobin detected in Nun-3 (electronic supplementary material, table S2) may derive from the dog host, especially as haemoglobin assigned to Carnivora was also identified in the sample. However, this protein could also have originated from caribou or seal, which has been reported from zooarchaeological evidence at Nunalleq (electronic supplementary material, table S11) [63]. Likewise, keratins assigned to Pecora (electronic

supplementary material, table S3) may have originated from caribou hide or intestines consumed by the dog that deposited the Nun-3 sample, but can be difficult to authenticate.

A second major bias relates to the database-matching approach, whereby the incompleteness of available databases and the lack of suitable reference proteomes introduce severe constraints on the ability to identify dietary sources as well as the gut microbiome and potential parasites in complex substrates such as palaeofaeces. If peptides cannot be accurately assigned because particular classes of proteins and species are under-represented in the databases, information may be lost or results misleading. As such it is important to make informed and careful decisions on the composition of databases compiled or selected in data analysis strategies, in order to be aware of potential taxonomic biases based on LC-MS/MS data.

The electronic supplementary material, table S11 presents an overview of taxa identified in zooarchaeological assemblages at Nunalleq [63] and the number of protein accessions available for each taxa in Swiss-Prot and UniProtKB (Swiss-Prot and TrEMBL combined). It is clear that variation in this representation has potential downstream effects on interpretation; for example we note that walrus, which has been reported as dog food in northern Alaska [17], is represented by greater than 29 500 proteins in UniProt, but only five are represented in Swiss-Prot. Caribou, which has been suggested as a major resource at Nunalleq is not represented in either database. Likewise, seals, another potential dietary source, are also under-represented, like many marine mammals. In an ideal world, a wide taxonomic diversity would be included in the search strategy; however, databases of large sizes can pose significant constraints [64]. These challenges are well documented in modern metaproteomics [65], but in palaeo-metaproteomics, they are compounded by the need to widen the search space to account for non-tryptic peptides and post-translational modifications resulting from degradation across archaeological timescales.

### (ii) Potential and future applications of faecal palaeoproteomics

We also note a number of topics which will have to be addressed by future research. For instance, methodological investigations are needed to understand the survival of proteins in palaeofaeces. Nunalleq is recognized for its exceptional biomolecular preservation owing to permafrost conditions, but faeces is a highly degradable material, and it is uncertain how proteins preserve in samples that are older or from different archaeological contexts [33].

Future studies exploring the palaeoproteomic profiles of the GI tract may have the potential to reveal insights into GI physiology and disease [66]. For example, proteomic profiles in modern stool samples have shown to be distinct in dogs [67] and humans [68] with acute and chronic GI diseases such as inflammatory bowel disease. Faecal proteomics is also used in diagnosing colorectal cancer [69], celiac disease and cystic fibrosis [70], and quantitative methods may be useful for investigating the expression of immune proteins in palaeofaeces as a sign of infection. Such analyses can inform about the health status of humans and animals, and may help researchers track the emergence and prevalence of GI diseases in the past.

Finally palaeoproteomic analyses of palaeofaeces may offer new insight into community-level gut microbiome composition and function [71], which is complementary to

ancient metagenomic approaches [35]. The gut microbiome is otherwise only preserved in mummified remains, and information from palaeofaeces is therefore a valuable contribution to the understanding of microbiome evolution. The gut microbiome is known to impact overall physiological health through a wide range of functions including digestion, defence against pathogens and immune system response [72]. It responds to changes in diet and shows marked differences between wild and captive mammals [73]. Such analyses could be complicated in dogs; however, as they are known to sometimes engage in coprophagy, while ethnographic accounts from northern Alaska indicate that dog diet may consist of caribou and human faeces [17]. While here we have applied a protein extraction which deliberately reduced the bacterial content, future studies may adopt alternative extractions where this microbiological component can be uncovered, in order to reveal insights into the composition and function of ancient gut microbiomes.

## 4. Conclusion

Through the analysis of dog coprolites from the site of Nunalleq, Alaska, we demonstrate that palaeofaeces is a viable substrate for palaeoproteomic analysis. We recovered a range of caniform proteins associated with pancreatic juice, small intestine, colon, rectum and stomach and show that the preserved proteome profile is reflective of a host digestive proteome. We further demonstrate that palaeofaeces retains direct dietary information that is accessible through metaproteomic analysis, which, complemented by ZooMS on preserved bone fragments within the palaeofaeces, provides novel insights into tissues and species consumed. Our results demonstrate that dogs at Nunalleq, a pre-contact North American Arctic site, consumed a broad selection of salmonid species available in the local environment, including coho, chinook, sockeye and chum, as well as bones from other canids. Our study also highlights several current challenges in the reconstruction of diet through palaeoproteomic analysis of paleofeces and coprolites, including the need for a better understanding of protein preservation in these substrates, as well as the development of more comprehensive reference databases to maximize protein identifications. Optimized methods to characterize host, dietary and microbial protein sources within coprolites and paleofeces have the potential to provide novel insight into human and non-human animal diet, GI physiology and disease and microbiome composition and function.

Data accessibility. The mass spectrometry proteomics data have been deposited in the ProteomeXchange Consortium (dataset identifier PXD025714) via the MassIVE partner repository (http://massive. ucsd.edu; MSV000087339). ZooMS data have been deposited in Zenodo: https://doi.org/10.5281/zenodo.4290988.

Authors' contributions. AK.W.R.: conceptualization, data curation, formal analysis, investigation, methodology and writing—original draft; J.H.: formal analysis, investigation, supervision, writing—original draft, writing—review and editing; K.KR.: formal analysis, methodology, writing—review and editing; E.M.-M.: conceptualization, resources, writing—review and editing; K.B.: conceptualization, resources, writing—review and editing; M.M.: formal analysis, methodology, writing—review and editing; K.M.: formal analysis, writing—review and editing; M.C.: conceptualization, validation, writing—review and editing; E.C.: conceptualization, supervision, writing—review and editing; C.S.: conceptualization, methodology, supervision, writing—review and editing

All authors gave final approval for publication and agreed to be held accountable for the work performed therein.

Competing interests. The authors declare no competing interests.

Funding. This project has received funding from the European Union's EU Framework Programme for Research and Innovation Horizon 2020 under Grant Agreement no. 676154 and the Danish National Research Foundation award PROTEIOS (DNRF128). The onsite collection of samples was carried out by staff and students from the University of Aberdeen, volunteer excavators and the residents of Quinhagak. Fieldwork/primary sampling was funded by an AHRC (AH/K006029/1) grant awarded to Rick Knecht (Aberdeen), K.B. and Charlotta Hillerdal (Aberdeen) and an AHRC-LabEx award (AH/N504543/1) to K.B., Rick Knecht (Aberdeen), Keith Dobney (Liverpool) and Isabelle Sidéra (Nanterre). We had logistical and planning support for fieldwork by Qanirtuuq Incorporated, Quinhagak, Alaska, and the people of Quinhagak, who we also thank for sampling permissions. We thank Prof. Jesper Velgaard Olsen at the Novo Nordisk Center for Protein Research for providing access and resources, which was also funded in part by a donation from the Novo Nordisk Foundation (grant no. NNF14CC0001). Additional support was provided through the New Frontiers in Research grant no. (SSHRC NFRFE-2018-00066).

Acknowledgements. We further acknowledge Takumi Tsutaya for valuable insight in implementing the protein extraction protocol and subsequent discussions, and Samantha Preslee for completing the ZooMS extractions and MALDI-TOF analysis in conjunction with the Technology Facility at the University of York. We are grateful to our three anonymous referees for their constructive comments and suggestions on our original manuscript.

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
