## [Peer Review File · Proceedings of the Royal Society B: Biological Sciences]

Review History

RSPB-2021-0020.R0 (Original submission)

Review form: Reviewer 1

Recommendation

Accept with minor revision (please list in comments)

Scientific importance: Is the manuscript an original and important contribution to its field?

Excellent

General interest: Is the paper of sufficient general interest?

Excellent

Quality of the paper: Is the overall quality of the paper suitable?

Excellent

Is the length of the paper justified?

Yes

Should the paper be seen by a specialist statistical reviewer?

No

Do you have any concerns about statistical analyses in this paper? If so, please specify them explicitly in your report.

No

It is a condition of publication that authors make their supporting data, code and materials available - either as supplementary material or hosted in an external repository. Please rate, if applicable, the supporting data on the following criteria.

Is it accessible?

Yes

Is it clear?

Yes

Is it adequate?

Yes

Do you have any ethical concerns with this paper?

No

Comments to the Author

This is a fascinating, exhaustively performed and clear study of dog faeces. I am impressed with the attention to detail, especially with regard to the limitations of the approach, and this is an excellent springboard that will no doubt launch dozens of similar studies. I have only a handful of comments.

Abstract - I was disappointed not see more detail here with regard to the specific findings. Many people may not get past the abstract and it's important to tell the entire story of the paper, including which dietary sources were found, here.

I'm aware of a new publication regarding the history of the first dogs in the Americas which may make for a better citation in the introduction (<https://doi.org/10.1073/pnas.2010083118>)

Line 186. Many non-Artic dogs do not possess an increase in AMY2B copy number and the association between starch and the increase in this locus is also suspect. I understand this is a minor point, but I would be a bit more cautious about how this gene is characterised.

Line 260. This is an excellent discussion of limitations of the approach and the call for better reference datasets is welcome and important.

Conclusions. This section is a bit brief and I feel the authors have missed an opportunity to go beyond the recap they offer and instead discuss the larger ramifications and future potential of this approach more generally.

Review form: Reviewer 2

Recommendation

Accept with minor revision (please list in comments)

Scientific importance: Is the manuscript an original and important contribution to its field?

Excellent

General interest: Is the paper of sufficient general interest?

Excellent

Quality of the paper: Is the overall quality of the paper suitable?

Excellent

Is the length of the paper justified?

Yes

Should the paper be seen by a specialist statistical reviewer?

No

Do you have any concerns about statistical analyses in this paper? If so, please specify them explicitly in your report.

No

It is a condition of publication that authors make their supporting data, code and materials available - either as supplementary material or hosted in an external repository. Please rate, if applicable, the supporting data on the following criteria.

Is it accessible?

Yes

Is it clear?

Yes

Is it adequate?

Yes

Do you have any ethical concerns with this paper?

No

Comments to the Author

This is such an interesting study that incorporates paleoproteomics and ZooMS in a novel way that has a lot of promise for future research in archaeology and paleontology. The sample size is a bit small, but I understand that destructive sampling and expensive analyses need to be limited in most studies of this kind and it opens the door to future research. Overall, the experimental design is very well planned and the manuscript is well-written and clear. The findings of both host and dietary proteomes is very exciting for the field of paleoproteomics. The manuscript is of really high quality and I only have a few comments/questions.

Did you do any additional morphological analyses on nun-5 to definitively exclude it from being a coprolite, such as sectioning? Is there any evidence of other coprolites from this debris area? Do you have any hypotheses for what it may be if it's not a coprolite?

Did you find any additional evidence of bacteria in the other samples (nun-1-4)?

This is a bit tangential, but after BLASTing one of the peptides from nun-5, these bacteria came up which is really interesting given the environment your samples were collected:
<https://www.ncbi.nlm.nih.gov/pmc/articles/PMC4921121/>

In Table S3, you only have nun 1-3 listed. Did you not detect keratins in samples 4 or 5?

Line 156: Please be specific about what collagens were found.

Review form: Reviewer 3

Recommendation

Major revision is needed (please make suggestions in comments)

Scientific importance: Is the manuscript an original and important contribution to its field?

Good

General interest: Is the paper of sufficient general interest?

Excellent

Quality of the paper: Is the overall quality of the paper suitable?

Good

Is the length of the paper justified?

Yes

Should the paper be seen by a specialist statistical reviewer?

No

Do you have any concerns about statistical analyses in this paper? If so, please specify them explicitly in your report.

Yes

It is a condition of publication that authors make their supporting data, code and materials available - either as supplementary material or hosted in an external repository. Please rate, if applicable, the supporting data on the following criteria.

Is it accessible?

Yes

Is it clear?

Yes

Is it adequate?

Yes

Do you have any ethical concerns with this paper?

No

Comments to the Author

In this manuscript, the authors perform proteomic analyses of archaeological dog feces, providing information on the diets of arctic dogs that cohabitated with humans. I have no expertise as regards our current knowledge of dog domestication and its intersection with human culture and practices; thus, I can't comment on this aspect of the manuscript and will limit my comments to the paleoproteomics. Paleoproteomic analyses of feces is an interesting topic, and I think this paper is appropriate for Proc B. However, there are some major points that must be addressed before I can recommend it for publication.

First, the parameters for the database searching performed here are very permissive, considering the data was obtained with a Q-Exactive HF-X, and in my opinion should be rerun with more rigorous thresholds. Specifically, the fragment error tolerance was set to 0.07 Da, when 0.02 Da would be more appropriate. Additionally, the FDR should be set to 1%, as is becoming standard

for paleoproteomics. In this current manuscript it is at 5%.

Second, although the authors do acknowledge and discuss the difficulty with distinguishing between “host” and “diet” proteins, there are at least a couple of examples of proteins that the authors identify “digestive” that in my opinion are more parsimoniously referred to “diet.” Specifically, authors identify chondroadherin and collagen (Page 4, Line 156) as ECM proteins that like arose from the intestines (page 5, Line 182). Although these proteins are present in those tissues, they are also very common in bone. And, given the fact that the authors state that dogs bones at the site show evidence of butchery, and bone fragments from other dogs were even found in one fecal sample (Page 6, Line 246) – the later direct evidence that the dog(s) at this site were eating other dogs – the assignment of these proteins to the host instead of its meal seems dubious. Thus, I think there needs to be a more robust discussion of how the distinction between eater/eaten has been made that addresses the evidence for the host having consumed other canids. The current referral of collagen I to the host simply because it is canid (Page 5, Line 182) is not sufficient.

Third, the use of ZooMS to identify the species of bone fragments present in a few feces specimens is, somewhat, a misapplication of ZooMS. The value of ZooMS is for high-throughput, broad identification of a large number of samples in a short amount of time. The way this manuscript is written, it seems to suggest that peptide mass finger printing is somehow higher-resolution for species identification. E.g., (Page 7, Line 258) “We also confirm that palaeoproteomics and ZooMS can provide high resolution insight into short term dietary intake”; and (Page 6, Line 217). “Although this protein is used to obtain information at species level in ZooMS analysis, the coverage obtained in the LC-MS/MS dataset was not sufficient to assign these to a level below Salmoninae.”

In fact, LC-MS/MS is capable of providing a much more robust and higher resolution identification of proteins than ZooMS. The reason that the collagen coverage was poor in LC-MS/MS in comparison is likely that those samples were prepared for protein recovery from feces, not bone, which is a biomineralized material and requires demineralization before protein solubilization with GuHCl. This is also probably why the collagen recovery from “Method A,” which used acid demineralization, was better than Method B, which did not (and not because the bones had been through a digestive tract, as the authors speculate on Page 6, Line 230). In short, LC-MS/MS could have more robustly identified these bones fragments, and given that there were only 5 specimens and not 100, it is unclear why the authors switched to a less rigorous method to ID the bone.

Decision letter (RSPB-2021-0020.R0)

26-Feb-2021

Dear Dr Speller:

Your manuscript has now been peer reviewed and the reviews have been assessed by an Associate Editor. The reviewers’ comments (not including confidential comments to the Editor) and the comments from the Associate Editor are included at the end of this email for your reference. As you will see, the reviewers and the Editors have raised some concerns with your manuscript and we would like to invite you to revise your manuscript to address them.

We do not allow multiple rounds of revision so we urge you to make every effort to fully address all of the comments at this stage. If deemed necessary by the Associate Editor, your manuscript will be sent back to one or more of the original reviewers for assessment. If the original reviewers

are not available we may invite new reviewers. Please note that we cannot guarantee eventual acceptance of your manuscript at this stage.

Research ethics:

Use of animals and field studies:

It is a condition of publication that you make available the data and research materials supporting the results in the article. Please see our Data Sharing Policies (<https://royalsociety.org/journals/authors/author-guidelines/#data>). Datasets should be deposited in an appropriate publicly available repository and details of the associated accession number, link or DOI to the datasets must be included in the Data Accessibility section of the article (<https://royalsociety.org/journals/ethics-policies/data-sharing-mining/>). Reference(s) to datasets should also be included in the reference list of the article with DOIs (where available).

Please submit a copy of your revised paper within three weeks. If we do not hear from you within this time your manuscript will be rejected. If you are unable to meet this deadline please let us know as soon as possible, as we may be able to grant a short extension.

Best wishes,

Dr John Hutchinson, Editor

Associate Editor

Comments to Author:

Thank you for submitting your research article, which has now been reviewed by three expert reviewers. I agree with their assessment that the data reported are interesting, that this clever application of paleoproteomics has the potential to make important contributions to our understanding of canine domestication processes, and the manuscript is well written. While all reviews are supportive of this research, one reviewer has substantive concerns about the statistical and analytical approaches. I encourage the authors to carefully address the points raised. The authors should also engage thoughtfully with the other requests for revision, in particular with the requests to expand aspects of the discussion.

Reviewer(s)' Comments to Author:

Referee: 1

Comments to the Author(s)

This is a fascinating, exhaustively performed and clear study of dog faeces. I am impressed with the attention to detail, especially with regard to the limitations of the approach, and this is an excellent springboard that will no doubt launch dozens of similar studies. I have only a handful of comments.

Abstract - I was disappointed not see more detail here with regard to the specific findings. Many people may not get past the abstract and it's important to tell the entire story of the paper, including which dietary sources were found, here.

I'm aware of a new publication regarding the history of the first dogs in the Americas which may make for a better citation in the introduction (<https://doi.org/10.1073/pnas.2010083118>)

Line 186. Many non-Artic dogs do not possess an increase in AMY2B copy number and the association between starch and the increase in this locus is also suspect. I understand this is a minor point, but I would be a bit more cautious about how this gene is characterised.

Line 260. This is an excellent discussion of limitations of the approach and the call for better reference datasets is welcome and important.

Conclusions. This section is a bit brief and I feel the authors have missed an opportunity to go beyond the recap they offer and instead discuss the larger ramifications and future potential of this approach more generally.

Referee: 2

Comments to the Author(s)

This is such an interesting study that incorporates paleoproteomics and ZooMS in a novel way that has a lot of promise for future research in archaeology and paleontology. The sample size is a bit small, but I understand that destructive sampling and expensive analyses need to be limited in most studies of this kind and it opens the door to future research. Overall, the experimental design is very well planned and the manuscript is well-written and clear. The findings of both host and dietary proteomes is very exciting for the field of paleoproteomics. The manuscript is of really high quality and I only have a few comments/questions.

Did you do any additional morphological analyses on nun-5 to definitively exclude it from being a coprolite, such as sectioning? Is there any evidence of other coprolites from this debris area? Do you have any hypotheses for what it may be if it's not a coprolite?

Did you find any additional evidence of bacteria in the other samples (nun-1-4)?

This is a bit tangential, but after BLASTing one of the peptides from nun-5, these bacteria came up which is really interesting given the environment your samples were collected:
<https://www.ncbi.nlm.nih.gov/pmc/articles/PMC4921121/>

In Table S3, you only have nun 1-3 listed. Did you not detect keratins in samples 4 or 5?

Line 156: Please be specific about what collagens were found.

Referee: 3

Comments to the Author(s)

In this manuscript, the authors perform proteomic analyses of archaeological dog feces, providing information on the diets of arctic dogs that cohabitated with humans. I have no expertise as regards our current knowledge of dog domestication and its intersection with human culture and practices; thus, I can't comment on this aspect of the manuscript and will limit my comments to the paleoproteomics. Paleoproteomic analyses of feces is an interesting topic, and I think this paper is appropriate for Proc B. However, there are some major points that must be addressed before I can recommend it for publication.

First, the parameters for the database searching performed here are very permissive, considering the data was obtained with a Q-Exactive HF-X, and in my opinion should be rerun with more rigorous thresholds. Specifically, the fragment error tolerance was set to 0.07 Da, when 0.02 Da would be more appropriate. Additionally, the FDR should be set to 1%, as is becoming standard for paleoproteomics. In this current manuscript it is at 5%.

Second, although the authors do acknowledge and discuss the difficulty with distinguishing between "host" and "diet" proteins, there are at least a couple of examples of proteins that the authors identify "digestive" that in my opinion are more parsimoniously referred to "diet." Specifically, authors identify chondroadherin and collagen (Page 4, Line 156) as ECM proteins that like arose from the intestines (page 5, Line 182). Although these proteins are present in those tissues, they are also very common in bone. And, given the fact that the authors state that dogs

bones at the site show evidence of butchery, and bone fragments from other dogs were even found in one fecal sample (Page 6, Line 246) – the later direct evidence that the dog(s) at this site were eating other dogs – the assignment of these proteins to the host instead of its meal seems dubious. Thus, I think there needs to be a more robust discussion of how the distinction between eater/eaten has been made that addresses the evidence for the host having consumed other canids. The current referral of collagen I to the host simply because it is canid (Page 5, Line 182) is not sufficient.

Third, the use of ZooMS to identify the species of bone fragments present in a few feces specimens is, somewhat, a misapplication of ZooMS. The value of ZooMS is for high-throughput, broad identification of a large number of samples in a short amount of time. The way this manuscript is written, it seems to suggest that peptide mass finger printing is somehow higher-resolution for species identification. E.g., (Page 7, Line 258) “We also confirm that palaeoproteomics and ZooMS can provide high resolution insight into short term dietary intake”; and (Page 6, Line 217). “Although this protein is used to obtain information at species level in ZooMS analysis, the coverage obtained in the LC-MS/MS dataset was not sufficient to assign these to a level below Salmoninae.”

In fact, LC-MS/MS is capable of providing a much more robust and higher resolution identification of proteins than ZooMS. The reason that the collagen coverage was poor in LC-MS/MS in comparison is likely that those samples were prepared for protein recovery from feces, not bone, which is a biomineralized material and requires demineralization before protein solubilization with GuHCl. This is also probably why the collagen recovery from “Method A,” which used acid demineralization, was better than Method B, which did not (and not because the bones had been through a digestive tract, as the authors speculate on Page 6, Line 230). In short, LC-MS/MS could have more robustly identified these bones fragments, and given that there were only 5 specimens and not 100, it is unclear why the authors switched to a less rigorous method to ID the bone.

Author's Response to Decision Letter for (RSPB-2021-0020.R0)

See Appendix A.

RSPB-2021-0020.R1 (Revision)

Review form: Reviewer 1

Recommendation

Accept as is

Scientific importance: Is the manuscript an original and important contribution to its field?

Good

General interest: Is the paper of sufficient general interest?

Excellent

Quality of the paper: Is the overall quality of the paper suitable?

Excellent

Is the length of the paper justified?

Yes

Should the paper be seen by a specialist statistical reviewer?

No

Do you have any concerns about statistical analyses in this paper? If so, please specify them explicitly in your report.

No

It is a condition of publication that authors make their supporting data, code and materials available - either as supplementary material or hosted in an external repository. Please rate, if applicable, the supporting data on the following criteria.

Is it accessible?

Yes

Is it clear?

Yes

Is it adequate?

Yes

Do you have any ethical concerns with this paper?

No

Comments to the Author

I'm satisfied with the changes the authors have made in response to my comments on their original draft, and recommend publication.

Decision letter (RSPB-2021-0020.R1)

10-Jun-2021

Dear Dr Speller

I am pleased to inform you that your manuscript entitled "Palaeoproteomic analyses of dog palaeofaeces reveal a preserved dietary and host digestive proteome" has been accepted for publication in Proceedings B. Congratulations!!

Data Accessibility section

Open Access

Paper charges

Sincerely,

Dr John Hutchinson

Appendix A

We are grateful to the three referees for their helpful comments on the manuscript, and we have outlined our specific responses to their comments and queries in blue font below.

Referee: 1

Comments to the Author(s)

This is a fascinating, exhaustively performed and clear study of dog faeces. I am impressed with the attention to detail, especially with regard to the limitations of the approach, and this is an excellent springboard that will no doubt launch dozens of similar studies. I have only a handful of comments.

Abstract – I was disappointed not see more detail here with regard to the specific findings. Many people may not get past the abstract and it's important to tell the entire story of the paper, including which dietary sources were found, here.

Good point, we have updated the abstract to include the dietary source identified, within the 200 word limit.

I'm aware of a new publication regarding the history of the first dogs in the Americas which may make for a better citation in the introduction (<https://doi.org/10.1073/pnas.2010083118>).

Thank you, we have included this reference in the introduction.

Line 186. Many non-Artic dogs do not possess an increase in AMY2B copy number and the association between starch and the increase in this locus is also suspect. I understand this is a minor point, but I would be a bit more cautious about how this gene is characterised.

Good point - we have updated our statement to "Differences in protein expression between dogs and humans have been observed including in saliva [52], while variation may also exist between different dog lineages. For example, some non-Arctic dogs display increased copy numbers of the AMY2B gene responsible for the production of amylase, the primary enzyme involved in starch digestion [53]."

Line 260. This is an excellent discussion of limitations of the approach and the call for better reference datasets is welcome and important.

Thank you.

Conclusions. This section is a bit brief and I feel the authors have missed an opportunity to go beyond the recap they offer and instead discuss the larger ramifications and future potential of this approach more generally.

We have updated the conclusion with further details, including on future methodological developments and the potential of new insight into human and non-human animal diet, gastrointestinal physiology and disease and microbiome composition and function.

Referee: 2

Comments to the Author(s)

This is such an interesting study that incorporates paleoproteomics and ZooMS in a novel way that has a lot of promise for future research in archaeology and paleontology. The sample size is a bit small, but I understand that destructive sampling and expensive analyses need to be limited in most studies of this kind and it opens the door to future research. Overall, the experimental design is very well planned and the manuscript is well-written and clear. The findings of both host and dietary proteomes is very exciting for the field of paleoproteomics. The manuscript is of really high quality and I only have a few comments/questions.

Did you do any additional morphological analyses on nun-5 to definitively exclude it from being a coprolite, such as sectioning? Is there any evidence of other coprolites from this debris area? Do you have any hypotheses for what it may be if it's not a coprolite?

We did not do additional morphological analyses on Nun-5. Although there are other coprolites from this area, coprolite morphology can vary significantly, and morphological characteristics can be further modified by compression or fragmentation (e.g., Shillito et al. 2020. *Earth-Science Reviews*, 103196.). It's possible that Nun-5 may represent another type of semi-mineralized organic material, however, the proteomic data could not provide any additional information on its origin.

Did you find any additional evidence of bacteria in the other samples (nun-1-4)?

The protein extraction method applied in this study was specifically designed to remove bacteria, however, following the reanalysis of the data, a few bacterial proteins were detected. These include Major outer membrane lipoprotein Lpp 1 in Nun-2 and Outer membrane protein A in Nun-4, both assigned to Enterobacteriales. In addition, 50S ribosomal protein L7/L12 assigned to Actinobacteria was recovered from Nun-3. Finally, Nun-5 contained 60 kDa chaperonin deriving from Burkholderiales. An earlier pilot study using a different extraction method was done on two of the samples, and this did recover more bacterial peptides, but was not optimal for the recovery of host or dietary proteins. Clearly additional studies testing the effectiveness of different extraction techniques are required to optimize methods for host/dietary and bacterial proteomes (noted in Discussion, pg 8).

This is a bit tangential, but after BLASTing one of the peptides from nun-5, these bacteria came up which is really interesting given the environment your samples were collected:

<https://www.ncbi.nlm.nih.gov/pmc/articles/PMC4921121/>

This is indeed an interesting observation for this peptide, however we note that this peptide also matches with 100% identity to other proteobacteria, and therefore its taxonomic specificity cannot be confidently ascertained beyond this phylum level.

In Table S3, you only have nun 1-3 listed. Did you not detect keratins in samples 4 or 5?

We did detect keratins in Nun-4 and Nun-5, however these were also found in the extraction blanks control and therefore were listed in Table S4 as contaminants. We have updated the title of Table S3 to make this more explicit. These are retained in Table S3 as they could represent endogenous identifications. This is discussed in a section of the manuscript.

Line 156: Please be specific about what collagens were found.

We've updated the manuscript to note that Collagen type I was identified (alpha-1(I) and alpha 2(I) chains).

Referee: 3

Comments to the Author(s)

In this manuscript, the authors perform proteomic analyses of archaeological dog feces, providing information on the diets of arctic dogs that cohabitated with humans.

I have no expertise as regards our current knowledge of dog domestication and its intersection with human culture and practices; thus, I can't comment on this aspect of the manuscript and will limit my comments to the paleoproteomics.

Paleoproteomic analyses of feces is an interesting topic, and I think this paper is appropriate for Proc B. However, there are some major points that must be addressed before I can recommend it for publication.

First, the parameters for the database searching performed here are very permissive, considering the data was obtained with a Q-Exactive HF-X, and in my opinion should be rerun with more rigorous thresholds. Specifically, the fragment error tolerance was set to 0.07 Da, when 0.02 Da would be more appropriate. Additionally, the FDR should be set to 1%, as is becoming standard for paleoproteomics. In this current manuscript it is at 5%.

The reviewer raises an excellent point. We welcomed the opportunity to re-analysed our data with 0.02 Da fragment error tolerance and with a correction to a 1% FDR, using a more recent version of Swiss-Prot (release date: 2020_02). We have updated the manuscript, tables, and figures with this revised dataset. This re-analysis has resulted in some minor adjustments to our identified proteins, but do not significantly change the results or the implications of the paper.

Second, although the authors do acknowledge and discuss the difficulty with distinguishing between "host" and "diet" proteins, there are at least a couple of examples of proteins that the authors identify "digestive" that in my opinion are more parsimoniously referred to "diet." Specifically, authors identify chondroadherin and collagen (Page 4, Line 156) as ECM proteins that like arose from the intestines (page 5, Line 182). Although these proteins are present in those tissues, they are also very common in bone. And, given the fact that the authors state that dogs bones at the site show evidence of butchery, and bone fragments from other dogs were even found in one fecal sample (Page 6, Line 246)—the later direct evidence that the dog(s) at this site were eating other dogs—the assignment of these proteins to the host instead of its meal seems dubious. Thus, I think there needs to be a more robust discussion of how the distinction between eater/eaten has been made that addresses the evidence for the host having consumed other canids. The current referral of collagen I to the host simply because it is canid (Page 5, Line 182) is not sufficient.

The reviewer raises a good point here. Interestingly, the canid bone fragment was recovered from Nun-4, while canid collagens and chondroadherin were recovered from Nun-3. Nevertheless, we've updated the section on Dietary Proteins (pg 7) to note that some putative host proteins may derive from dietary sources.

Third, the use of ZooMS to identify the species of bone fragments present in a few feces specimens is, somewhat, a misapplication of ZooMS. The value of ZooMS is for high-throughput, broad identification of a large number of samples in a short amount of time. The way this manuscript is written, it seems to suggest that peptide mass fingerprinting is somehow higher-resolution for species identification.

E.g., (Page 7, Line 258) “We also confirm that palaeoproteomics and ZooMS can provide high resolution insight into short term dietary intake”; and (Page 6, Line 217). “Although this protein is used to obtain information at species level in ZooMS analysis, the coverage obtained in the LC-MS/MS dataset was not sufficient to assign these to a level below Salmoninae.”

In fact, LC-MS/MS is capable of providing a much more robust and higher resolution identification of proteins than ZooMS.

We agree with the reviewer that ZooMS is frequently used for high-throughput screening, however, this is not the only way in which it is utilized in the published literature. ZooMS can often offer sufficient taxonomic resolution with greatly reduced analysis cost, data analysis time, and processing power than LC-MS/MS and therefore has been used reliably in many studies even on a small number of samples. We agree that LC-MS/MS is capable of producing high-resolution identifications when conducted on the bone material. However, as there are species-specific ZooMS markers available for Pacific salmon (Korzow-Richter et al. 2020), we opted to attempt this rapid, cost-effective approach ahead of additional LC-MS/MS analysis on the individual bones.

The manuscript does not compare ZooMS and LC-MS/MS analyses on the same material (the LC-MS/MS analysis is on a sample of palaeofaeces and the ZooMS is on bone fragments found within the palaeofaeces) so there is no real comparison for species-specific resolution. To clarify that we are making no direct comparisons between the methods we have updated Page 6 Line 217 “Although type I collagen can be used to obtain species level information through ZooMS and/or LC-MS/MS analysis of bone material, the coverage obtained in the palaeofaeces LC-MS/MS dataset was not sufficient to assign these to a level below Salmoninae”.

The reason that the collagen coverage was poor in LC-MS/MS in comparison is likely that those samples were prepared for protein recovery from feces, not bone, which is a biomineralized material and requires demineralization before protein solubilization with GuHCl. This is also probably why the collagen recovery from “Method A,” which used acid demineralization, was better than Method B, which did not (and not because the bones had been through a digestive tract, as the authors speculate on Page 6, Line 230). In short, LC-MS/MS could have more robustly identified these bones fragments, and given that there were only 5 specimens and not 100, it is unclear why the authors switched to a less rigorous method to ID the bone.

We agree that collagen coverage was poor in LC-MS/MS because the extraction method was optimized for feces/coprolites rather than bone and did not include a demineralization step. We

also agree that this is why Method A produced superior results to method B, since it included a demineralization step which releases a greater fraction of the preserved collagen in the bone. We have clarified this point in our discussion.

While for some taxa, LC-MS/MS can provide a more precise taxonomic identification than ZooMS, analyses by Richter et al. 2020 have demonstrated that the level of taxonomic resolution capable for Pacific salmon species identification is the same for LC-MS/MS and ZooMS— i.e, to the species level. As such, ZooMS is equally robust as a taxonomic method in this instance.

We have also added a statement in our results section noting that while for Pacific salmon, ZooMS is sufficient for species level identifications, this LC-MS/MS may be required for other taxa (pg 6). *“These results further demonstrate that ZooMS analyses can successfully be applied to semi-digested bone fragments which are frequently preserved in dog and human palaeofaeces [58] and may provide complementary insight into diet [42]. While ZooMS can provide species-level information for Pacific salmonids, higher-resolution analyses, such as LC-MS/MS may be required for this level of taxonomic resolution for mammal species [59].*